# Benzodiazepine Boom: Tracking Etizolam, Pyrazolam, and Flubromazepam from Pre-UK Psychoactive Act 2016 to Present Using Analytical and Social Listening Techniques

**DOI:** 10.3390/pharmacy12010013

**Published:** 2024-01-12

**Authors:** Anthony Mullin, Mark Scott, Giorgia Vaccaro, Giuseppe Floresta, Davide Arillotta, Valeria Catalani, John M. Corkery, Jacqueline L. Stair, Fabrizio Schifano, Amira Guirguis

**Affiliations:** 1Psychopharmacology, Drug Misuse & Novel Psychoactive Substances Research Unit, University of Hertfordshire, Hatfield AL10 9AB, UK; anthony.mullin@hotmail.co.uk (A.M.); m.scott6@herts.ac.uk (M.S.); g.vacarro@herts.ac.uk (G.V.); giuseppe.floresta@unict.it (G.F.); v.catalani@herts.ac.uk (V.C.); j.corkery@herts.ac.uk (J.M.C.); j.stair@herts.ac.uk (J.L.S.); f.schifano@herts.ac.uk (F.S.); 2Department of Drug and Health Sciences, University of Catania, 95131 Catania, Italy; 3School of Clinical Pharmacology and Toxicology, University of Florence, 50121 Florence, Italy; davide.arillotta@yahoo.it; 4Pharmacy, Medical School, The Grove Extension, Swansea University, Swansea SA2 8PP, UK

**Keywords:** designer benzodiazepines, social media, poly-drug use, analytical characterisation

## Abstract

Introduction: The designer benzodiazepine (DBZD) market continues to expand whilst evading regulatory controls. The widespread adoption of social media by pro-drug use communities encourages positive discussions around DBZD use/misuse, driving demand. This research addresses the evolution of three popular DBZDs, etizolam (E), flubromazepam (F), and pyrazolam (P), available on the drug market for over a decade, comparing the quantitative chemical analyses of tablet samples, purchased from the internet prior to the implementation of the Psychoactive Substances Act UK 2016, with the thematic netnographic analyses of social media content. Method: Drug samples were purchased from the internet in early 2016. The characterisation of all drug batches were performed using UHPLC-MS and supported with ^1^H NMR. In addition, netnographic studies across the platforms X (formerly Twitter) and Reddit, between 2016–2023, were conducted. The latter was supported by both manual and artificial intelligence (AI)-driven thematic analyses, using numerous.ai and ChatGPT, of social media threads and discussions. Results: UHPLC-MS confirmed the expected drug in every sample, showing remarkable inter/intra batch variability across all batches (E = 13.8 ± 0.6 to 24.7 ± 0.9 mg; F = 4.0 ± 0.2 to 23.5 ± 0.8 mg; P = 5.2 ± 0.2 to 11.5 ± 0.4 mg). ^1^H NMR could not confirm etizolam as a lone compound in any etizolam batch. Thematic analyses showed etizolam dominated social media discussions (59% of all posts), with 24.2% of posts involving sale/purchase and 17.8% detailing new administration trends/poly-drug use scenarios. Artificial intelligence confirmed three of the top five trends identified manually. Conclusions: Purity variability identified across all tested samples emphasises the increased potential health risks associated with DBZD consumption. We propose the global DBZD market is exacerbated by surface web social media discussions, recorded across X and Reddit. Despite the appearance of newer analogues, these three DBZDs remain prevalent and popularised. Reporting themes on harm/effects and new developments in poly-drug use trends, demand for DBZDs continues to grow, despite their potent nature and potential risk to life. It is proposed that greater controls and constant live monitoring of social media user content is warranted to drive active regulation strategies and targeted, effective, harm reduction strategies.

## 1. Introduction

Benzodiazepines (BZDs), first introduced as pharmaceuticals in 1960 by Hoffman-La Roche, have become widely prescribed sedative, hypnotic, anti-convulsant, and muscle relaxant medicines. They exert their central nervous system (CNS) depressant effects by enhancing gamma-aminobutyric acid (GABA) actions on the GABA type A (GABA_A_) receptor, resulting in an increased influx of chloride ions across the receptor channels. ’Designer’ BZDs (DBZDs) emerged in the early-2000s as ’legal’ alternatives to controlled BZDs in Europe [1]. Their market has experienced a significant resurgence, with a thriving underground economy that continually adapts to evade regulatory constraints, raising public health concerns [1,2]. Drug-related deaths (DRDs) continue to increase globally, presenting increasingly complex drug scenarios where BZDs are key contributors to the cause of deaths [3,4]. In Scotland, for example, deaths from BZDs, including diazepam and etizolam, increased from 26% in 2008 to 57% in 2022 of the total number of drug misuse deaths [5,6]. The implications of these BZD analogues’ DRDs, accidental or otherwise, are widely debated [7,8]. Often, BZD DRDs are related to the combined inhibitory effects of both BZDs on GABA_A_ receptors, and other CNS depressants, such as opioids, on opioid receptors in respiratory control centres, thus facilitating respiratory depression when co-consumed, and potentially leading to deaths [9,10,11]. The risks are further amplified by the uncertainty surrounding the increasing potencies of DBZD analogues, dosages, and the high prevalence of counterfeit tablets masquerading as prescribed BZDs like diazepam and alprazolam. These fake psychoactive substances, typically manufactured by organised criminal groups, often include active adulterants, cutting agents, and/or by-products resulting from illicit processes [12,13,14]. Adulterants in counterfeit drugs are a significant concern; they have the potential to enhance both the recreational and harmful effects of the substance. Moreover, these adulterants undergo changes over time as a tactic to evade scrutiny and control by health services or food regulations [15].

Among the multitude of designer BZDs, three analogues - etizolam, pyrazolam, and flubromazepam - emerged as prominent players before the implementation of the United Kingdom’s (UK) Psychoactive Substances Act (PSA) in 2016 [16,17]. These substances were readily available for purchase online on the surface web, and marketed as ’legal highs’ [18,19]. During this pre-regulatory era, the scientific community seized an opportunity to track these BZDs, which are not internationally controlled, using advanced analytical techniques such as High-Performance Liquid Chromatography (HPLC) and Nuclear Magnetic Resonance (NMR) spectroscopy [19,20,21]. The objective was to comprehensively characterise these compounds, shedding light on their chemical composition and structural properties. However, the implementation of the PSA in May 2016 [17] brought significant changes to the landscape of designer BZDs in the UK. The Act sought to curb the proliferation of ’legal highs’ by imposing stringent regulations on demand for psychoactive substances [22]. In this transformed regulatory environment, the availability of etizolam, pyrazolam, flubromazepam, and similar compounds was restricted, raising questions about their continued presence in the street and online markets [23].

Social listening has proven invaluable in proactively monitoring evolving trends, market dynamics, and potential risks associated with DBZD use [24,25]. It allows the continuous surveillance of internet and social media platforms frequented by people who use drugs (PWUD), enabling the prompt identification and prediction of emerging trends, health hazards, and misuse patterns. In this study, X and Reddit were selected due to their vast user bases, popularity, and relevance. X has 397 million users, with 206 million daily active users, while Reddit boasts 430 million active users and 52 million daily interactions [26,27,28]. Both platforms offer open-topic discussions, user anonymity, and free expression within certain limits. They support file attachments, web links, and multiple languages. Their broad range of topics and robust user engagement policies make them conducive for anonymous observation and social studies [24,29]. Innovative web crawler technology (NPS*finder*^®^) has also been employed to monitor openly shared content on the surface web by PWUD and drug enthusiasts, leading to the discovery of novel psychoactive substances (NPS) previously unreported by international bodies [25]. This approach highlights the power of social media in advancing research on drug trends.

This research aimed to explore the evolutionary trajectory of three DBZDs—etizolam, pyrazolam, and flubromazepam—from their pre-regulation prominence to the present day. To achieve this aim, our research design incorporated a multifaceted approach to extract valuable insights from emerging drug markets. It encompassed a comprehensive quantitative analytical assessment of known drug samples and complemented this with a thorough netnographic analysis of trends within the PWUD community. This combined approach aimed to recognise significant changes across various drug use communities and groups and identify emerging trends that may pose substantial health risks, allowing for real-time responses. Our investigation involved identifying the predominant themes of online discussions using artificial intelligence (AI). This information enables a deeper understanding of the historical and current online presence of these popular NPS drugs and specific trends in information-sharing content.

## 2. Methods

### 2.1. Forensic Analyses

Using UHPLC-MS as the primary technique, complemented by ^1^H NMR, this study reports on street sample purity, active drug ingredient (ADI) content, and identified chemical structure of three tablets from each batch (*n* = 15) of street samples, purchased before the implementation of the PSA 2016.

#### 2.1.1. Reference Materials and Reagents

Analytical reference standards were utilised in this study, with diazepam serving as the internal standard, procured from Sigma Aldrich (Merck Life Sciences UK Ltd., Gillingham, UK). Etizolam, flubromazepam, and pyrazolam were obtained from Chiron AS (Trondheim, Norway). Deuterated methanol-d4 was acquired from Cambridge Laboratories (Cambridge MA, USA). UHPLC-grade methanol (MeOH) and formic acid 98–100% were sourced from Fisher Scientific (Loughborough, UK). The in-house production of Millipore Water was carried out using a Millipore 0.22 µm Filtration System from Merck Life Sciences UK Ltd. (Gillingham, UK). The Millipake Express 20 filter system was employed, ensuring Water Quality at 15.2 MΩ·cm at 25 °C.

#### 2.1.2. Drug Samples

The batches of pyrazolam (*n* = 4), Flubromazepam (*n* = 5), and etizolam (*n* = 6) tablets were purchased over the open internet from three separate vendors (Figure 1). From each batch, 3 tablet samples were randomly selected for analysis. Prior to analytical testing, each of the selected tablets were weighed, using a Mettler Toledo Balance (0.01 mg–220 g), operated inside an enclosed safety cabinet (BIGNEAT F3-XIT). All analytical procedures remained in line with both the ICH Harmonised Guideline for Validation of Analytical Procedures [30] and the Analytical Procedures and Method Validation for Drugs and Biologics guidelines [31] and were recorded in line with our UK Home Office agreement. Samples were also weighed before and after UHPLC-MS and NMR analyses, to record the evidence of experimental/analytical/sample transfer loss, using the same Mettler Toledo Balance. All preparations and experimental analyses were performed at an ambient room temperature of ca. 26 °C.

#### 2.1.3. UHPLC-MS

The experimental procedure employed Ultra-High-Pressure Liquid Chromatography–Mass Spectrometry with Diode Array Detector (UHPLC-MS-DAD) utilising a Waters Acquity CM system (UHPLC-MS), equipped with a Waters Autosampler, Acquity PDA Diode Array Detector, Waters Acquity Qda Mass Spectrometer, and MassLynx V4.2 SCN976 software for system operation and data processing. A Phenomenex Kinetex C18 100 Å Column, with dimensions of 100 × 2.1 mm inner diameter and 2.6 µm pore size, was procured from Phenomenex in Macclesfield, UK. The mobile phases consisted of A (0.1% formic acid in water) and B (0.1% formic acid in methanol), both filtered using a Sinter Filter system (0.22 µm). For sample analysis, a gradient method was optimised, and it was initiated with a 50:50 A:B ratio over 0.5 min (min), followed by a linear gradient of 10:90 from 0.5 to 3.5 min, and finally equilibration to 50:50 at 4.0 min, running until complete with a 6 min total run time. The column temperature was maintained at 28 °C. Sample injection volume was 0.2 µL, performed in triplicate and interspersed with double-blank injections of MeOH between all samples. Spectral data for all samples were generated using a Waters PDA at a rate of 20 sampling points per second, set to 1.2 nm resolution across the wavelength range of 190–400 nm. Electrospray (ESI) at approximately 3–5 kV resulted in ions contained in aerosol droplets, which were protonated and detected in the form [MH]+ in positive-ion mode. This process facilitated the identification of fragments, enabling the determination of the molecule’s identity.

#### 2.1.4. ^1^H NMR

Proton (^1^H) NMR spectroscopy was conducted using a JEOL ECA600 spectrometer featuring an HCN probe, and spectral analysis was carried out using Delta 5.3.1 software. The standard acquisition parameters were set as follows: X_width 6.25 [us], X_acquisition time 1.45 s, X_angle 45°, relaxation delay 4 sec, and 64 scans. Signal assignment was accomplished through a comparative analysis with the known spectra of reference standards.

#### 2.1.5. Sample Analysis

All MS and NMR spectra acquired from the samples were initially cross-referenced with verified reference materials. Additional electronic resources used for reference included the Perkin Elmer online predictive NMR resource Beaconsfield, UK. Further comparisons were made utilising SWGDRUG online resources (Woodbridge, VA, USA), Cayman Chemical online resources (Ann Arbor, MI, USA), NPSDiscovery.org (Willow Grove, PA, USA), J Wiley & Sons (Medford, MA, USA), nmrdb.org (Cali, Colombia), MOLBASE online resources (Shanghai, China), Mass Bank of North America online resources (Davis, CA, USA), Royal Society of Chemistry ChemSpider online resources (Cambridge, UK), National Library of Medicine online resources (Bethesda, MD, USA), and Ceondo GmbH Chemical online resources (Gelsenkirchen, Germany).

Reference and sample preparations using both UHPLC-MS and ^1^H NMR analyses are detailed in Appendix A.

### 2.2. Netnographic Methods

A comprehensive netnographic analysis of discussions, related to these three drug analogues, was conducted between 1 January 2016 to 18 May 2023.

#### 2.2.1. Primary Data Search

Data were retrieved, stored, and handled in accordance with published userbase privacy policies, guidance for data distribution, and active pharmaceutical ingredient (API) developer agreements, related to both media platforms [32,33]. All X and Reddit posts were retrieved using their own proprietary active API advanced search function, with access to information from X dating back to 2006, but with no time limits from Reddit. Inclusion and exclusion criteria of both Reddit and X posts are shown in Table 1.
X Search: Using the X search function, several broad date range operators (1 January–31 December 2016–2022 and 1 January–15 May 2023) were applied, using a single search term, in each case the name of the drug compound, i.e., etizolam, flubromazepam, or pyrazolam. Additional filtering criteria were applied to include tweets (X posts) in English and exclude re-tweets.Reddit Search: Using the Reddit search function, single search terms were applied, in each case the name of the drug compound, i.e., etizolam, flubromazepam, or pyrazolam. Every thread identified through the applied searches were reviewed, and all relevant posts recorded within the target date range (1 January–31 December 2016 up to 15 May 2023) were included for analysis; posts falling outside of the timeframe were discarded. Additional filtering criteria, including duplicates analysis and thread re-posts, were applied manually, using Microsoft 365 Excel (version no. 2311, 2023).

Each bank of data from both X and Reddit searches were extracted to a Word document, using Microsoft Office 365, where personal information, contact details, and telephone numbers were redacted, and all remaining information was formatted into date ranges, in accordance with both X and Reddit user and data-handling policies, as well as the study’s ethics framework [27,28,32,33].

#### 2.2.2. Data Cleaning

All data across both platforms were initially cleaned manually. Each Tweet and Reddit post was read manually and checked against the experimental inclusion and exclusion criteria, and, where required, any external links posted were followed to their endpoint to determine the content and appropriateness of the post. This process was implemented by AM and randomly checked by AG to exclude researcher bias and mitigate selection bias. Any duplicate posts and retweets which bypassed the user search interface were identified and removed. Any post that did not contain any illicit drug-related activity or information but included the name of the specific drug in either the post body or within the title was retained; these non-illicit activities included music to listen when using the drug, a music festival or nightclub event, and the name of artwork. After cleaning, the data sets were grouped by year.

#### 2.2.3. Data Analyses

Data set analyses were performed using two different methods: a manual analysis of themes and media content using Microsoft Excel 2023 and an AI-driven data and thematic analysis using the platforms numerous.ai [34] and ChatGPT [35]. The AI method was employed to reduce researcher bias in conducting the manual thematic analysis. Alternate formatting was required for both platforms, due to restrictions on character analysis and output and numerous.ai being a plug-in for spreadsheet analyses [36].
Manual Discussion/Theme Analysis: Each post was assessed for keywords and identified themes. Any key words/origination sources matching the experimental inclusion criteria were recorded for quantitative purposes. Any content found to bridge multiple topics was included in each associated theme.Numerous.ai Analysis. All cleaned data sets were exported to Google sheets for analysis within their annual grouping, for example, 1 January 2017 to 31 December 2017. Due to the program’s restrictions on the numbers of characters per data set, in terms of both input and output [36], it was necessary to further group the annual data sets into subsets of 10. Using a blank cell adjacent to the data, the following command prompt was typed into that cell, [=ai (“i want you to act as an expert in qualitative content analysis and analyse this post for me. Identify all the themes and then present them in bullet points. Please also consider any potential biases or contextual factors that may impact your analysis”)], followed by the manual selection of the cell subset range within the spreadsheet to be analysed. Each set of numerous.ai responses was captured, cut and pasted into a Microsoft 365 Word (Version no. 2311, 2023) document in preparation for ChatGPT analysis.ChatGPT Analysis: Analytical functionality on ChatGPT is less restricted in terms of character input and output; therefore, using ChatGPT version 3.5, we were able to group all numerous.ai responses, per drug, into three individual drug-specific Microsoft Word (version no. 2311. 2023) documents for analysis. To process the numerous.ai responses, we typed the following input into each Word document: [“could you resume the 5 most frequent themes/biases among these; and rank these is order or prevalence?”]. This resulted in the five most common themes and biases being identified by the application of this dual-AI method. The responses were then collated using Microsoft 365 Word (version no. 2023), for final assessment and inclusions.

## 3. Results

### 3.1. UHPLC-MS

The purpose of the analytical investigation was to identify the drugs of interest contained within the purchased illicit samples against data attained from reference standard analysis. Once identified, any inter/intra batch purity variations and tested tablet sample purities were quantitatively assessed using an internal standard method. All results were compared to the published literature. A remarkable purity variability per tested batch is reported (Table 2) with results ranging between 13.8 ± 0.6 mg and 24.7 ± 0.9 mg (E), 4.0 ± 0.2 mg and 23.5 ± 0.8 mg (F), and 5.4 ± 0.2 mg to 11.5 ± 0.4 mg (P). This represents an inter/intra batch variations of 8.2 to 29.2%, 2.4 to 24.6%, and 5.2 to 12.0%, for etizolam, flubromazepam, and pyrazolam, respectively. Packaging labels were presented for each batch, showing the proposed concentration of active drug ingredient (ADI) in each tablet, with etizolam and pyrazolam batches containing 1 mg and flubromazepam 8 mg of ADI per tablet. Variability between expected drug content and quantitative analysis showed a broad range of drug content variation within the tested samples, giving a final sample purity range between 53 and 2450% of the expected ADI content.

#### 3.1.1. Etizolam Reference Standard and ET1-6 Tablets

UHPLC-MS comparison analysis between the analytical-grade etizolam reference standard and drug samples confirmed the presence of etizolam (4-(2-chlorophenyl)-2-ethyl-9-methyl-6H-thieno[3,2-f][1,2,4]triazolo[4,3-a][1,4]diazepine) in all samples/batches (Figure 2), showing a single response at 2.45 min, with the internal standard, diazepam, eluting at 2.73 min. The ESI spectra for all samples showed molecular ions at *m*/*z* 343 and 345, along with a visible 3:1 ratio between the reported ions, therefore indicating a 37-Cl isotope related to the chlorine atom attached to the thiophene ring, associated with the etizolam chemical structure (detailed analysis can be found in Appendix A). The comparison of etizolam tablets shows high inter-batch and sample purity variability. For example, etizolam tablet sample 5, analysed in triple-triplicate batches 1–3 (ET5 B1-3), showed the lowest concentration of all tablet samples in batch 2, with an average drug content of 13.8 ± 0.6 mg. Sample ET1 B2 showed the highest purity at 24.7 ± 0.9 mg. Whilst samples ET5 B1-3 resulted in the lowest concentration range of all etizolam batches, ranging from 13.8 ± 0.6 to 15.5 ± 0.6 mg; the highest concentration cluster of sample batches was found within sample ET4 B1-3, ranging between 23.6 ± 1.4 and 24.7 ± 0.8 mg. At the point of purchase, the suggested purity for each batch of tablets was proposed as 1 mg of the etizolam ADI; as a minimum, these tested samples showed a maximum increase in purity of 2450% when compared with the proposed drug content (Table 3).

#### 3.1.2. Flubromazepam Reference Standard and FT1-5 Tablets

The UHPLC-MS analysis of the flubromazepam reference material (Figure 3) shows a single peak elution at 2.45 min, with the internal standard addition, diazepam, eluting at 2.68 min (Table 4). Both peak elution profiles were visible across every tablet sample (FT1–FT5 (B1-3)), with the elution time concordant with the reference standard spectra. Both lone peaks were further compared against the literature [39] to confirm the flubromazepam molecule. ESI spectra for all samples showed protonated molecular ions at *m*/*z* 333 and 335 (Table 4), corresponding with the average molecular mass at 333.15 g/moL^−1^. The DAD response showed a λmax around 268 nm, typically within the region expected for a BZD derivative, with some overlap expected from the diazepam compound. We confidently confirmed the presence of flubromazepam (7-bromo-5-(2-fluorophenyl)-1,3-dihydro-2*H*-1,4-benzodiazepin-2-one) in all samples/batches (Figure 3). The comparison of flubromazepam tablets shows a remarkable inter-batch and sample purity variability. Flubromazepam tablet sample 1, analysed in triple-triplicate batches 1-3 (FT1 B1-3), showed the lowest concentration of all tablet samples in batch 2, at 4.0 ± 0.2 mg. Sample FT5 B2 resulted in the highest purity, 23.5 ± 0.8 mg. Samples FT1 B1-3 resulted in the lowest concentration range of all flubromazepam batches, ranging from 4.0 ± 0.2 mg to 4.5 ± 0.2 mg, and the highest concentration cluster of samples was found within sample FT5 B1-3, ranging from 22.7 ± 0.7 mg to 23.5 ± 0.8 mg. The largest inter-batch variation is found within FT2 B1-3, varying between 14.2 ± 0.6 mg and 22.2 ± 0.7 mg. At the point of purchase, the suggested purity for each batch of tablets was proposed as 8 mg of the flubromazepam ADI; as a minimum, these tested samples show a maximum increase in overstated content of 204% when compared with the advertised drug content (Table 3).

#### 3.1.3. Pyrazolam Reference Standard and PT1-4 Tablets

The UHPLC-MS comparison analysis of the pyrazolam reference material and drug samples confirmed the presence of pyrazolam (8-bromo-1-methyl-6-pyridin-2-yl-4H-[1,2,4]triazolo[4,3-a][1,4]benzodiazepine) in all samples/batches (Figure 4), showing a single peak for pyrazolam at 1.02 min, with the internal standard, diazepam, eluting at 2.68 min (Table 5). ESI spectra for all samples showed protonated molecular ions at m/z 354 and 356. Pyrazolam, with a computed average mass of isotopic variants of 354.21 g/mol, was identified from parent ions of m/z 353 and 355, both from protonated adducts and from sodium adducts.. The comparison of pyrazolam tablet samples indicated a high inter-batch and sample purity variability. Pyrazolam tablet sample 1, analysed in triple-triplicate batches 1-3 (FT1 B1-3), showed the lowest concentration of all tablet samples in batch 1, at 5.4 ± 0.2 mg. Sample FT4 B2 resulted in the highest purity, 11.5 ± 0.4 mg. Samples FT1 B1-3 resulted in the lowest concentration range of all pyrazolam batches, ranging from 5.4 ± 0.2 mg to 8.2 ± 0.3 mg, and the highest concentration range within pyrazolam batches identified within samples FT4 B1-3, ranging from 10.5 ± 0.3 mg to 11.5 ± 0.4 mg. The largest inter-batch variation is found within PT2 B1-3, varying between 7.1 ± 0.3 mg and 11.1 ± 0.4 mg. At the point of purchase the suggested purity for each batch of tablets was proposed as 1 mg of the pyrazolam ADI; as a minimum, across all tested samples and batches, this represents a maximum increase in purity of 1090% when compared with the stated drug content (Table 3).

### 3.2. ^1^H NMR Discussion

The confirmation of the appropriate BZD was considered achieved if all the peaks seen in the reference sample were also seen in the drug sample with the correct number of protons per peak (Appendix A). After integrating drug sample spectra, any peaks with a large integral value (>10) were considered excipients and ignored for comparison purposes. The ^1^H NMR for flubromazepam and pyrazolam reference standards showed peaks that corresponded to the drug structures. However, for etizolam, the reference material did not show any signals for the methylene protons in the seven-membered ring. Because of this, we can confirm that ^1^H NMR only confirmed the presence of flubromazepam and pyrazolam in their respective tablet samples.

#### 3.2.1. Etizolam

As expected, four protons from the aromatic ring were seen around 7.5 ppm, a single proton at 6.5 ppm from the thiophene ring together with a two-proton quartet at 2.85 ppm (from ethyl side chain), a three-proton singlet at 2.7 ppm from methyl group on azole ring, and a three-proton triplet at 1.3 ppm from the ethyl side chain (Table 2). As mentioned above, no peaks from the methylene protons of the seven-membered ring could be seen in the reference spectra. Across all drug samples, the methyl proton signal expected from the ethyl side chain could not be identified; although this may be a result of excipients obscuring the signal in the aliphatic region, around 1.3 ppm. Samples ET1, ET2, and ET3 showed peaks at 4.48 ppm and 4.34 ppm, which may possibly indicate methylene protons expected from the seven-membered ring. However, samples ET4, ET5, and ET6 did not show these peaks. Sample ET5 showed a weak noisy spectrum with only the aromatic protons being detected, whereas ET6 showed a spectrum comparable to ET4. Table 3 indicates that ET5 did have a lower average drug content than the other samples. Hence, it was not possible to confirm the presence of etizolam in any of the tested drug samples, using ^1^H NMR alone. The theoretical assessment of the etizolam chemical structure suggests that two methylene proton peaks should be expected, associated with the seven-membered ring. The SWGDRUG database shows a single broad peak at 4.9 ppm containing two protons; however, this was run in deuterated DMSO [42]. The standards and samples in this study were run in deuterated methanol, so possibly, the methylene protons were swamped by the water signal from the methanol, which was expected at around 4.8 ppm. The J Coupling reports for ET2, ET4 and ET6 NMR showed 0H at 3.18, 2.04, and 3.60 respectively. We recognise this as a common rounding function of the Delta J Software (version 5.3.1) when the program is not able to fully integrate a peak.

#### 3.2.2. Flubromazepam

Under ^1^H NMR analysis, all flubromazepam drug samples showed the same peak profiles evident from the flubromazepam reference standard (Table 2), confirming the presence of flubromazepam in the drug samples. However, we record that the NH proton signal was not identified in the reference material, which ought to be seen around 10 ppm. However, this is a result of the exchange between the sample and the MeOH-d4 solvent used for the analysis. All other peaks have been well documented [43], confirming the presence of flubromazepam in all samples, FT1-FT5.

#### 3.2.3. Pyrazolam

All drug samples showed the pyrazolam peaks seen in the reference pyrazolam proton spectrum (Table 2), confirming the presence of pyrazolam in all four of the tablet samples. J Coupling reports for PRS and PT1 show an OH at 1.90, due to unresolved peak integration (Figure 5).

### 3.3. Quantitative Netnographic Analysis

Primary searches using the X (T) and Reddit (R) internal search functions, over the period of 1st January 2016 to 16th May 2023, yielded *n* = 1593 posts (*Tn* = 997, *Rn* = 596). Post cleaning, relevant posts for inclusion were reduced to 1183 (*Tn* = 772, *Rn* = 411), with 457 posts falling into multiple themes. Year after year, X has appeared to be the most prevalent social media platform in terms of etizolam discussions, with substantially more posts (*n* = 704) than Reddit (*n* = 91), with flubromazepam and pyrazolam taking precedence across Reddit. Etizolam has shown prominence across X since 2016, although post-COVID Reddit has witnessed a rise in etizolam posts and discussions (2016–2021 (*n* = 0), 2022 (*n* = 41), 2023 (*n* = 80)). The total number of pyrazolam and flubromazepam posts across Reddit, across all years, are more prevalent than across X. After identifying the primary information source included in the posts, more PWUD have been shown to utilise Reddit over X (*Rn* = 406, *Tn* = 350). Information posted by drug testing organisations, educational establishments, police, and border force agencies, GOs, and NGOs vastly populate X compared to Reddit (*Rn* = 7, *Tn* = 422). Prior to the introduction of the PSA 2016, etizolam was the most prevalent drug in terms of social media posts during 2016 (*n* = 73), compared to flubromazepam (*n* = 4) and pyrazolam (*n* = 5). This trend has continued, up until 2023, with etizolam discussed the most, followed by pyrazolam as the second most prevalent drug, with flubromazepam being the least prevalent in discussions.

Manual netnographic analysis was undertaken methodically including the analysis of additional factors, such as post-origination, the inclusion of attachments, the investigation of attached links, and key words/phrases/basic themes forming a part of the discussion. Reddit was found to be populated with a higher number of discussions led by PWUD (*n* = 406), compared to X (*n* = 350). Across the three drugs chosen for analysis, etizolam discussions were clearly more prevalent in discussions led by PWUD (*n* = 406) across Reddit and were more populated by organisationally generated posts from the various organisations (*n* = 422). Both platforms showed a similarly high level of discussions around poly-drug/substance use on X, more highly populated by discussion around harms (*n* = 320), when compared with Reddit (*n* = 75). The high numbers of discussions around purchasing and selling drugs were found across both platforms (*Tn* = 165, *Rn* = 180). X posts contained the highest numbers of discussions around etizolam sales and harms, with 132 and 300 discussions, respectively. This represented a dramatic increase over the same discussion topics compared with Reddit. A dramatically more prevalent use of X than Reddit by the various organisations was also evident (*Tn* = 422, *Rn* = 7). Similarly, only X was recognised for posting illicit material seizure data (*n* = 43).

#### 3.3.1. Manual Qualitative and Quantitative Thematic Analysis

Manual analysis was conducted using keywords and manual content assessment to determine relevant themes. The responses were ranked to show the five most prevalent themes (Appendix A). The top three discussion themes were deemed to be harms, buying/selling, and effects, with 33, 29, and 24%, respectively, of the total posts across all platforms and all three drugs. The two remaining themes, other and trends, secured 21 and 17% of responses. The total number exceeds 100%, owing to overlapping discussions that were included on more than one theme.

#### 3.3.2. AI-Driven Qualitative and Quantitative Thematic Analysis

Automated thematic analysis conducted by two novel AI data analysis packages, numerous.ai and ChatGPT, was applied to the social media data. The original data were grouped into subsets of 10 data points and analysed using numerous.ai, resulting in 113 further subsets of detailed data analysis responses. Each new subset described multiple proposed themes associated with the original data sets. Each response was recorded and passed through ChatGPT for interpretive thematic analysis, ranked in the order of prevalence (Table 6).

The top three themes proposed by ChatGPT include drug use and experiences; harm reduction and advice; and safety concern and health effects, which were closely concordant with the themes identified by the manual analysis method. Manual analysis was reduced to single word thematic relevance; however, inclusion criteria and discussion-source analysis supported their direct comparison across the five automated themes. Examples are shown in Table 6, illustrating multiple social media entries that have driven thematic categorisation. The themes were ranked 1–5 based on their repeated occurrences. Some posts were included in more than one theme. As opposed to manual analysis, the automated responses were detailed and offered some insight. However, the crossover of information used as evidence to support any postulated theme was shown to be inefficient on occasions, and may be susceptible to bias as it focusses on a keyword in isolation from the overall context of the post. It is worth noting that AI analysis has limited the inclusion of certain terms such as ‘suicide’, ‘self-harm’ or offensive language.

## 4. Discussion and Conclusions

This study utilised a novel orthogonal research design to investigate the evolving landscape of three popularised DBZDs [44], etizolam, flubromazepam, and pyrazolam, to examine their presence within the illicit drug market and continued popularity within online communities, over the extended period of 2016 to 2023. Gold standard analytical methods for quantification were adopted, and the results were supported by both qualitative/quantitative netnographic analyses of social media posts (X and Reddit), and the novel application of artificial intelligence to contrast the manual thematic analysis. This study underscores the critical importance of engaging with social media platforms to identify and monitor current trends, some of which may elude regulatory authorities. Furthermore, it highlights the significance of continuous surveillance and analysis in informing public health policies and interventions designed to address emerging drug trends effectively.

All three BZDs are Class C drugs under the Misuse of Drugs Act 1971 and Schedule 1 drugs under the Misuse of Drugs Regulations 2001. Etizolam was first reported by the EMCDDA in 2011 and became internationally controlled in 2017 [45,46], prior to its inclusion in Schedule IV of the 1971 Convention on Psychotropic Substances, in 2020 at the Commission on Narcotic Drugs Sixty-third session [47]. It has consistently been identified by the WEDINOS drug checking laboratory, based in the UK (Welsh Emerging Drugs & Identification of Novel Substances) since 2013. It is a thienodiazepine analogue of triazolo 1,4-benzodiazepines and is structurally similar to fluclotizolam. It is a short-acting BZD and is claimed to act in a similar way to diazepam [12,48]. Its continued popularity is alarming, given the high prevalence of domestic manufacture in Scotland, in particular, its implications in drug-related deaths in the UK [49]. Conversely, flubromazepam is a long-acting BZD, claimed to act similarly to phenazepam, triazolam, and pyrazolam. It is structurally closer to phenazepam with a fluorine atom substituting for the chlorine atom in phenazepam [12,50]. Pyrazolam is a brominated triazolo BZD analogue that is structurally similar to alprazolam, but with a bromine atom substituting for the chlorine atom, and a pyridinyl group substituting for the phenyl group [18]. It is claimed to be 12 times more potent than diazepam [17,51] when the effects of 10 DBZDs were compared including the three BZDs of interest in this study [52]. It was found that flubromazepam was more potent, pyrazolam was the least potent but the most anxiolytic, and etizolam was the most euphoric among the three DBZDs. All three BZD batches were obtained in tablet forms as opposed to other studies which obtained pyrazolam and flubromazepam in pellet, powder, and blotter forms in 2014 and 2016 [44]. Surprisingly, the latter study could not obtain tablet forms of these BZDs [44].

Here, the empirical analyses of various etizolam tablet batches revealed that while quantification and mass spectrometry (MS) methods consistently identified etizolam in all the tablet batches, UHPLC-MS showed notable commonalities in terms of eluted peak areas across these batches [42,53]. This analysis exposed remarkable variations in purity both within and between batches, which is expected when dealing with illicit samples produced in clandestine manufacturing facilities. A detailed examination of the tablet batch analyses revealed specific omissions in each spectral response. Most notably, a deficiency of visible methyl protons in the ethyl side chain was a recurring pattern observed across all batches. Three of the samples, ET1, ET2, and ET3, did display visible proton peaks around 4.48 to 4.34 parts per million (ppm) but with one and two protons, respectively, rather than one each. Samples ET4, ET5, and ET6 did not show these two peaks. In all cases, the tablet samples did not show the methyl peaks from the ethyl side chain expected around 1.3 ppm possibly because of a large peak(s) from excipients in the tablets. In the case of sample ET4, resolution was poor, with only a few aromatic protons discernible. Consequently, it is concluded that confirmatory analysis through ^1^H NMR could not conclusively substantiate the presence of etizolam in any of the tablet samples when compared with the established literature and the reference standard [42].

Flubromazepam reference material was also analysed against the literature and spectral monographs, confirming MS fragmentation patterns [39,54,55]. With MS analysis, a second parent ion was identified at *m*/*z* 335 and was confirmed with the literature to correspond with the presence of 1 of 12 possible flubromazepam structural isotopes [43]. When comparing all tablet batches to their relevant reference spectra, there were significant visible differences in peak areas, confirming purity variations and clandestine operations to all flubromazepam tablet batches, equally. ^1^H NMR of the flubromazepam reference standard concurred with the published literature [39]. Each tablet batch was analysed against the reference standard and the known literature to confirm the presence of the flubromazepam structure within all tablet batches. However, the presence of NH protons, expected around 10.7 ppm, could not be identified; it can be confidently assumed that these peaks may be absent due to deuterium exchange with the deuterated methanol solvent. Given the confirmation of structural relevance achieved using UHPLC-MS, it can be assumed with confidence that flubromazepam was identified in all tested tablet batches.

Chromatographic peak retention times and MS fragmentation patterns from all tested pyrazolam tablet batches matched those of the reference standard. All data were compared to the published chromatographic monographs [41,56,57], to confirm the presence of pyrazolam within all tablet batches. Quantitative analysis confirmed significant inter/intra batch purity variability, although substantial differences in peak profile area was recorded across all batches; this adds to the probability that some quality processes have been lost during the manufacturing processes, as expected with clandestine compounds. UHPLC-MS spectral response across all pyrazolam tablet batches showed reduced peak areas, compared with the reference standard and the internal standard, diazepam. Considering that pyrazolam was confirmed, it seems sensible to assume that there may be some degree of signal quenching between the pyrazolam signal of unknown excipients, affecting the prominence of the target drug peak profile. This is indicative of street substances with unknown bulking agents. Given the internal standard correlation and calibration curve correlation remained within experimental limits, at correlation coefficient of R = 0.9999 or above, consistently, there is confidence in the quantitative process. The Proton NMR of the pyrazolam reference standard was confirmed with the published drug monographs [41]. All tested tablet batches were compared with the reference sample spectra and the literature to confirm the presence of pyrazolam, and the appropriate J Coupling report was presented for analysis. Similarly, there were some changes in spectral response, with sample PT2 and PT4 being more defined when compared with PT1 and PT3, thus concurring with the quantitative analysis.

The continuous monitoring of the three BZDs over the period 2016–2023 extended to social listening. Since 2020 and the impact of the COVID-19 pandemic, global reliance upon the internet has increased exponentially, with 5.16 billion internet users and 4.76 billion social media users accessing online information with high frequency [58]. The changing role of the internet has supported the evolving methods of obtaining DBZDs and facilitating their availability [59]. Although the BZDs evaluated in this study were obtained from the surface web pre-UK PSA (2016), novel marketing strategies allow their anonymous acquisition from the deep web including TOR (the Onion Routing Project) [59]. This article sheds light on the role of the internet in drug abuse, the dynamic nature of drug acquisition methods, and the challenges posed by the internet in regulating the accessibility of DBZDs [59,60]. Tracking online discussions enabled this study to gauge the historical and ongoing popularity and use of these three DBZDs across the various PWUD communities, and current findings are in line with previous research [22,23,24,61,62,63]. However, the significant variations in BZD samples purchased from online vendors is alarming, showing how the consumption roulette of these substances may play a key role in increasing drug-related deaths [64,65,66]. This research found that despite the known risks associated with DBZDs, especially etizolam, these drugs have remained highly popular and well revered amongst the online drug community. The latest WEDINOS annual report (2022/23) highlighted the prevalence of etizolam and the absence of both flubromazepam and pyrazolam, reporting etizolam as one of the most commonly identified drugs in samples collected from drug services, criminal justice settings, nighttime economy, healthcare providers, and other sources across the UK [67]. Often co-consumed with bromazolam and opioids [68], in this report, etizolam was the drug intended for purchase around 44 times [67,69,70].

Whilst the popularity of etizolam, across both X and Reddit, has substantially increased over time, online discussions regarding flubromazepam and pyrazolam appeared to decline remarkably across X, suggesting a community-shift in opinion surrounding these specific compounds. This may be due to the discussions across the community pertaining to overall effects, potency, effectiveness, availability, and adverse effects. However, both etizolam and flubromazepam were recently identified through a drug-checking service in the UK [70], blood samples from apprehended drivers and post-mortem from autopsies in Norway, and the analysis of blood tests related to individuals driving under the influence of drugs from 2017–2021 in the US [71].

Engagement with X and Reddit both increased dramatically during 2016–2023, with Reddit posts (*n* = 391) rising beyond X (*n* = 381). Interestingly, both platforms registered a dramatic shift in traffic origin; X became popular as an outlet for drug-related information posted by various organisations, including police, education, and health services (*n* = 422), compared with Reddit, which seem to be synonymous with PWUD communities, with low content engagement from organisations (*n* = 5). Across all drugs and the primary identified themes, the popularity of, and demand for, etizolam, by far outweighs that of pyrazolam and flubromazepam, even from the heydays of each drug around 2013 and 2012, respectively. This could possibly be due to its desired euphoric effect. Across all 1183 tweets and posts, trends including poly-drug use and new administration routes were dominant, primarily on X (*n* = 47), compared with Reddit (*Rn* = 24), with discussion around the combination of etizolam and Xanax, etizolam and heroin/methamphetamine, and etizolam and alcohol. Some reference was made to “smoking etiz using a vape”, but this was a lone example. Encompassing all new potential trends, Reddit appeared to receive the most posts from PWUD (*Rn* = 80, *Xn* = 49), with X being more of a conduit between organisations and the drug communities and users (*Xn* = 74, *Rn* = 1). The netnographic analysis recorded multiple shifts in what platforms are used, and by whom, for specific content; this may highlight a genuine need for the constant monitoring of social media content, to recognise and act upon changing trends and community demands, across this fluid marketplace.

The daily recommended dose for etizolam in a clinical setting is proposed between 1–4 mg daily [12,48]; however, up to 20 times this daily dose was found in the tested tablet batches in this study. Social media analysis shows a dramatically higher incidence of chatter around etizolam, with some users reporting, “I blacked out on Xanax and etizolam for three days–I don’t remember anything”, and “first time I took etizolam i lost a week of my life”, and, “bring back the etizolams and pyrazolams, bring them back!”. Yet, etizolam has witnessed a meteoric rise in popularity and usage since its launch in the illegal market in 2011. In 2019, it was linked to 59% of drug-related deaths in Scotland alone [72]. Etizolam has been shown to impair the ability to drive at low doses, and to exert serious toxicities at high doses [73]. In 2020, BZDs contributed to 73% of global drug-related deaths, with street BZDs, primarily street-bought etizolam, accounting for 66% of these fatalities in 2020 [74].

Comparisons with quantitative data reported by WEDINOS have provided some insight into the BZD purities identified in this study. WEDINOS have reported these three specific drugs within the top 10 prevalent substances sent in for analysis between 2014 and 2023, with etizolam being consistently reported up to 2023, flubromazepam up to 2021, and pyrazolam up to 2020 [65]. It is important to note that pyrazolam was not identified in WEDINOS annual reports in analysed samples, but as the BZD intended for purchase [70]. WEDINOS also reported the presence of etizolam in samples where the intent of purchase was another BZD such as alprazolam or diazepam. The concentrations of BZDs reported by WEDINOS ranges from 0.25 to 20 mg [70]. This is a cause of concern, not only because of the unknown strength of the BZD or the BZD analogue, which may be different from the label claim, but also from a pharmacokinetic perspective, where some of these BZDs may be short-, intermediate-, or long-acting drugs, with active metabolites [75,76]. In their 2021/22 report, etizolam was present in 22.6% of all tested samples purported to be diazepam, with 53.4% of those samples exhibiting etizolam as the lone active component [77].

Although each of these three substances continue to hold a strong presence in social media chatter, there are several new compounds superseding their predecessors’ positions. For example, in 2021, the EMCCDA reported the modification from flubromazepam to flubromazolam, a drug with a much higher potency due to the synthesis of triazolo derivatives within the flubromazepam compound. Reddit users reported a typical dose of flubromazepam ranging from 4 to 8 mg, whereas a typical dose of flubromazolam ranging from 0.25–0.5 mg, with flubromazepam being reported as less potent, less sedating, and more euphoric than flubromazolam, reportedly being a strong hypnotic and dysphoric molecule [78]. This higher potency would undoubtedly change the user experience and may have been a result of the original flubromazepam compound being unstable at ambient temperature [1]. It was also claimed that flubromazolam, which was not internationally controlled until 2020, was used to produce counterfeit diazepam, alprazolam, temazepam, and zopiclone [1,68,75].

Pyrazolam has been recorded as being sold in tablet form, with a concentration between 0.5 and 1 mg of active component per tablet. When considering the analytical data presented here, this is concerning, given the reported purity of between 5.4 ± 0.2 mg and 11.5 ± 0.4 mg per tablet. Reddit users reported that 0.5–0.75 mg is approximately equivalent to 10 mg of diazepam [79]. These findings reflect many concerns raised by the EMCDDA towards the increasing popularity of the manufacture of fake BZDs [1]. Netnographic thematic analysis showed here a positive correlation between the purity and effect variability amongst pyrazolam users, with some social media posts stating, for example, “Woke up and thought I would die-I over did the pyrazolam to sleep two days ago” and “a side effect of continual pyrazolam use was double vision, it’s such a strong benzo”, suggesting pyrazolam can be mis-dosed and even harmful. Despite DBZDs being heavily legislated against across the globe, these compounds remain highly sought after and available. All three BZDs were sought to self-medicate for insomnia and end a trip. Both etizolam and pyrazolam were sought to self-medicate for addiction and anxiety. Etizolam was also sought as a ‘downer’ and to manage craving from other drugs. This may suggest that current methods of predicting market and drug community developments, and how to deal with drug demands, are simply not effective enough. We acknowledge, in this study, that the tested samples were purchased prior to the introduction of the PSA (2016); yet in the more recent literature, the quantitative analysis of street purchased etizolam also suggests similar degrees of product purity and inter/intra batch variability. This demonstrates that DBZDs remain a serious cause of concerns, given the mortality statistics across Scotland and some European countries [80,81,82]. Thus, it can be inferred that either these drugs potentially remain under similar manufacturing controls as witnessed pre-2016, or there is a residual drug supply circulating among the various PWUD communities and supply chains.

These open web fora allow for the open promotion of positive drug-user opinions within the drug market, but they also provide a platform for positive harm reduction messages, harm reduction strategies, and evidence-based drug advice. However, this research highlighted that many posts and chat threads are unregulated and openly promote specific drug-taking behaviours, administration trends, poly-drug use, and poly-substance abuse, potentially encouraging replicative behaviours across the community [83]. This unrestricted discussion, compounded by the variability in drug purity and toxicodynamics/ toxicokinetics, poses significant risks, including harm and death as shown by the increasing numbers of BZD-related admissions to poison centres [84]. Laboratory analyses undertaken in this study also revealed inconsistencies in dosage and quality control, contributing to the unpredictability of these BZDs and the potential for harm. Despite well-documented risks, some BZDs remain popular among users. The current study highlighted the challenge faced by regulators and healthcare professionals in addressing the demand for BZDs and mitigating harm.

The DBZD market presents significant public health concerns due to its over-potent nature, variability, and continued popularity. Policymakers, regulatory agencies, and healthcare providers should urgently address these issues with effective harm reduction strategies, increased regulation, and targeted education to safeguard the well-being of individuals using these substances. Current findings underscore the need for immediate action in the face of this evolving challenge.

## 5. Limitations

The authors do recognise the several limitations of this study. Due to a range of restrictions, resulting from the COVID-19 pandemic in March 2020, we were unable to identify all adulterants across the tablet samples. In addition, the retrospective analysis of social media posts over a long period of time, using each platform’s proprietary search function, had limitations in terms of malleability and response accuracy. Although current research accurately represents what information can be found with each search, it is important to highlight that the main analysis themes were determined manually before any AI analysis was applied. AI’s performance was evaluated and compared to the manual analysis, and not as a substitute. Media searches and identified themes from the manual process were deliberately broad ranged, to gather the maximum amount of information within the specific timeframe. Another limitation to the current study is the adoption of just two social media sources, limiting the number of posts being assessed; this is because this is a novel approach employing combined methodologies and will be used to inform future research. We recognise that this initial analytical research was conducted using a team of researchers, strengthening analytical conclusions; however, the netnographical work was conducted by a lone researcher over a period of time. To mitigate any potential researcher bias, all findings were randomly verified by the senior researcher AG and discussed with a team, and themes were identified using a focus team to avoid thought singularity. At each stage, co-authors were consulted to validate the findings.

## Figures and Tables

**Figure 1 pharmacy-12-00013-f001:**
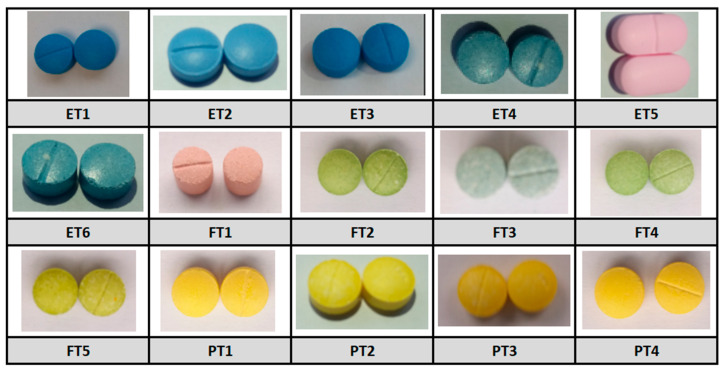
Tablet samples included in the study. NB: Each batch and tablet sample were coded for simplicity (etizolam tablets, ET1, ET2, ET3, ET4, ET5, ET6; flubromazepam tablets, FT1, FT2, FT3, FT4 and FT5; pyrazolam tablets, PT1, PT2, PT3 and PT4).

**Figure 2 pharmacy-12-00013-f002:**
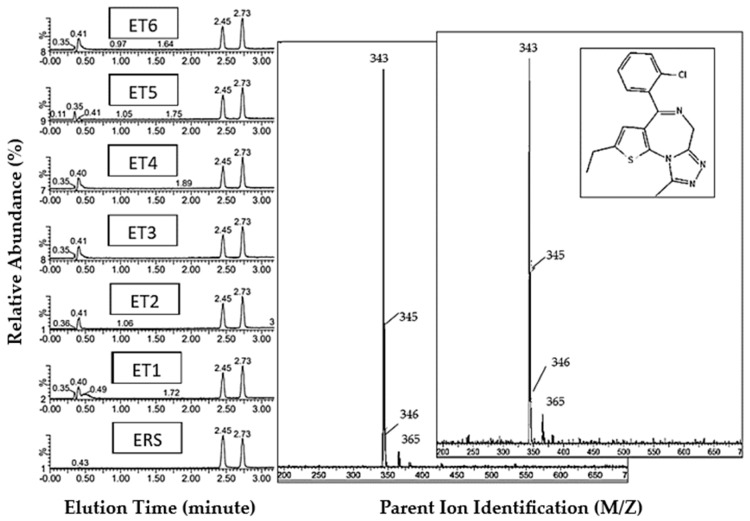
Etizolam UHPLC-MS analysis: peak elution for all etizolam tablet samples (ET) and etizolam reference standard (ERS). Chromatogram (left) shows etizolam mass elution at ca 2.45 min, with diazepam internal standard at ca 2.73 min; image shows sample descriptors. Reference standard mass spectra inlayed (middle), tablet sample ET1 mass spectra (right), and target molecule etizolam (inset). Corresponding to the average molecular mass for etizolam, accounting for Cl-35 and Cl-37 isotopes, we report a mass at 343.80 g/mol^−1^ [37]. Thus, protonated isotopic variants report parent ions for etizolam at *m*/*z* 343 and 345, respectively, which are identified in all spectra. Accounting for the presence of a sodium adducts MH+, a byproduct of the UHPLC-MS process, we identified a parent ion at 265 *m*/*z*, as expected [38].

**Figure 3 pharmacy-12-00013-f003:**
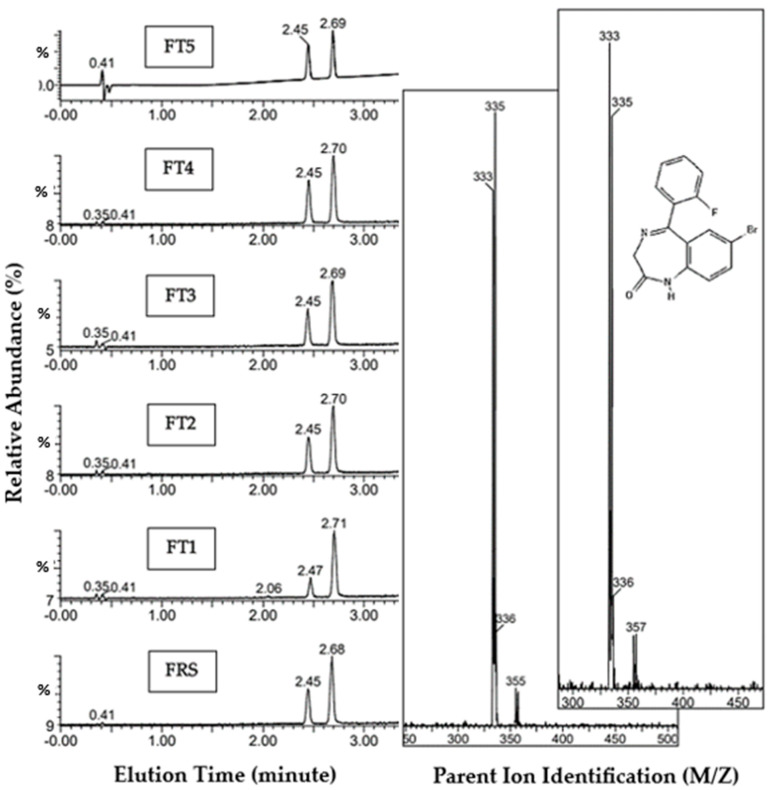
Flubromazepam UHPLC-MS analysis: peak elution for all flubromazepam tablet samples (FT) and flubromazepam reference standard (FRS). Chromatogram (**left**) show flubromazepam mass elution at ca. 2.45 min, with diazepam internal standard at ca. 2.68 min; image shows sample descriptors. Reference standard mass spectra inlayed (middle left (FRS), tablet sample FT1 mass spectra inlayed (middle right), and target molecule flubromazepam (inset). Flubromazepam molecular mass 333.150 g/mol^−1^ [39], with UHPLC-MS showing typical protonated ions in flubromazepam at *m*/*z* 333 and 335 [40].

**Figure 4 pharmacy-12-00013-f004:**
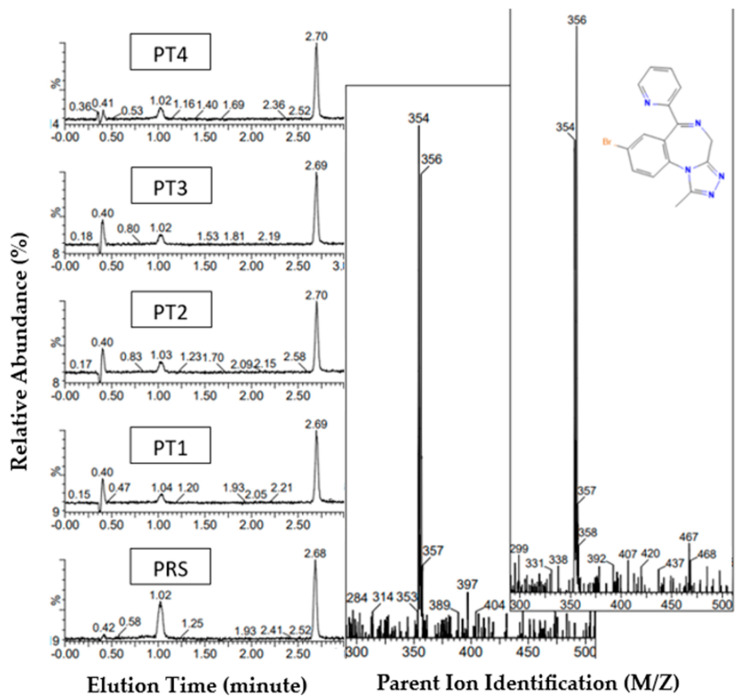
Pyrazolam UHPLC-MS analysis: peak elution for all pyrazolam tablet samples (PT) and pyrazolam reference standard (PRS). Chromatogram (**left**) show pyrazolam mass elution at ca. 1.02 min, with diazepam internal standard at ca. 2.68 min; image shows sample descriptors. Reference standard mass spectra inlayed (middle), tablet sample PT1 mass spectra (**right**), and target molecule pyrazolam (inset). Pyrazolam molecular mass 354.20 g/mol^−1^ [41]. MS shows a nominal mass of *m*/*z* 356, indicating the presence of Br-81 isotope protonated molecular ion.

**Figure 5 pharmacy-12-00013-f005:**
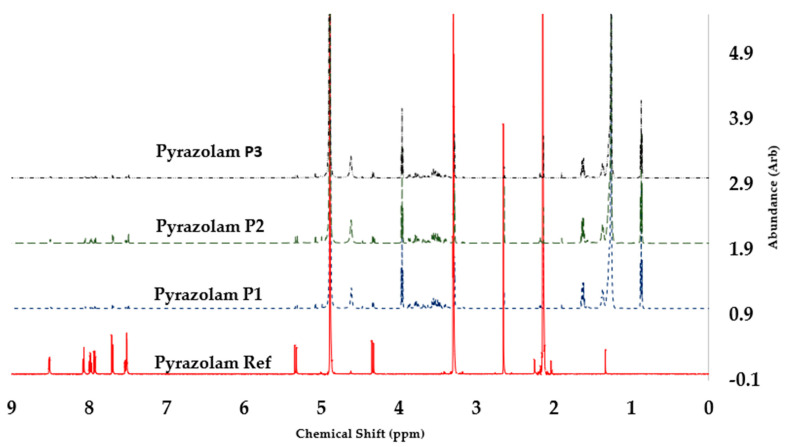
The 600 MHz ^1^H NMR comparison of pyrazolam tablet samples against pyrazolam reference standard, showing comparable spectral profiles, with sample P1 and P3 showing reduced profile definition attributable to drug concentration.

**Table 1 pharmacy-12-00013-t001:** X/Reddit API advanced search function parameters, showing key inclusion and exclusion criteria.

Search Terms	Inclusion Criteria	Exclusion Criteria
Etizolam, flubromazepam, and pyrazolam	Tweets and Reddit threads posted in English	Tweets and Reddit threads posted in another language other than English
	Keywords in discussion of named drugs under investigation, including polydrug use, polysubstance use, polyalcohol use, trends, and ingestion routes	Keywords and related terms not included in the discussion
	Keywords in discussion of drugs being used with/as a replacement for prescription drugs, any conversation around drug substitutions/new compounds	Re-tweets, Reddit reposts, and repetitions
	Keywords describing/discussing the supply/sale of the 3 drugs	Post falling outside the inclusion criteria
	Posts describing/discussing drug seizures, drug seizure information, seizure arrests/criminality surrounding the 3 drugs	Post falling outside the inclusion criteria

**Table 2 pharmacy-12-00013-t002:** Summary of the analytical findings for etizolam.

Drug Sample ID *	UHPLC-MS					^1^H-NMR (600 MHz, METHANOL-D4)	
	Drug Compound Retention Time (min)	Molecular Mass (*m*/*z*)	Base Peak (*m*/*z*)	Collection of Molecular Ions and Adducts	Confirmed ID * Using Literature (Cayman Chemical)	J Couplings Report	Confirmed ID * (Ref Std ** or Literature)
ERS	2.45	342.8	343	343, 345, 346	Etizolam	δ 7.52–7.42 (m, 4H), 6.47 (d, J = 1.4 Hz, 1H), 4.92–4.87 (m, 7H), 3.31–3.30 (m, 3H), 2.86–2.81 (m, 2H), 2.73–2.69 (m, 3H), 1.28 (t, J = 7.6 Hz, 3H)	Etizolam not confirmed
ET1	2.45	342.8	343	343, 345, 346, 365	Etizolam	δ 7.51–7.40 (m, 4H), 6.45 (s, 1H), 4.48 (d, J = 7.6 Hz, 1H), 4.34 (d, J = 7.6 Hz, 2H), 2.82 (q, J = 7.6 Hz, 2H), 2.69 (s, 3H), 2.25 (t, J = 7.6 Hz, 2H), 2.13 (s, 2H)	Etizolam not confirmed
ET2	2.45	342.8	343	343, 345, 346, 365	Etizolam	δ 7.51–7.40 (m, 4H), 6.45 (s, 1H), 4.48 (d, J = 7.6 Hz, 1H), 4.34 (d, J = 7.6 Hz, 2H), 3.18 (dd, J = 9.3, 7.9 Hz, 0H), 2.82 (q, J = 7.6 Hz, 2H), 2.69 (s, 3H), 2.25 (t, J = 7.6 Hz, 2H), 2.13 (s, 2H)	Etizolam not confirmed
ET3	2.45	342.8	343	343, 345, 346, 365	Etizolam	δ 7.51–7.40 (m, 4H), 6.45 (d, J = 1.4 Hz, 1H), 4.48 (d, J = 7.6 Hz, 1H), 4.34 (d, J = 7.6 Hz, 2H), 3.18 (dd, J = 9.3, 7.9 Hz, 1H), 2.84–2.81 (m, 2H), 2.69 (s, 3H), 2.25 (t, J = 7.2 Hz, 2H)	Etizolam not confirmed
ET4	2.45	342.8	343	343, 345, 346, 365	Etizolam	7.52–7.42 (m, 4H), 6.47 (s, 1H), 2.84 (dd, J = 7.6, 1.4 Hz, 2H), 2.71 (s, 3H), 2.26 (t, J = 7.6 Hz, 3H), 2.04 (s, 0H), 1.92 (s, 1H), 1.58 (d, J = 7.6 Hz, 2H), 0.89 (t, J = 6.9 Hz, 4H)	Etizolam not confirmed
ET5	2.45	342.8	343	343, 345, 346, 365, 367	Etizolam	δ 7.52–7.39 (m, 4H), 6.45 (s, 1H), 2.83 (q, 2H), 1.27 (s, 1H), 1.27–1.26 (m, 4H), 0.859–0.85 (2H)	Etizolam not confirmed
ET6	2.45	342.8	343	343, 345, 346, 365	Etizolam	δ 7.50–7.40 (m, 4H), 6.45 (s, 1H), 3.60 (d, J = 28.2 Hz, 0H), 2.83 (q, J = 7.6 Hz, 2H), 2.69 (s, 3H), 2.18–2.14 (m, 3H), 1.57 (t, J = 7.2 Hz, 2H), 0.87 (t, J = 7.2 Hz, 3H)	Etizolam not confirmed

* ID = identification; ** Ref Std = reference standard.

**Table 3 pharmacy-12-00013-t003:** Average drug concentrations of each batch of tablet, across triplicate working stock analysis.

Sample ID *	Average Drug Content Tablet (mg) with Standard Deviation (mg)	Sample ID *	Average Drug Content Tablet (mg) with Standard Deviation (mg)	Sample ID *	Average Drug Content Tablet (mg) with Standard Deviation (mg)
ET1 B1 R1-3	21.8 ± 0.8	ET6 B1 R1-3	21.7 ± 1.0	FT5 B1 R1-3	22.7 ± 0.7
ET1 B2 R1-3	24.7 ± 0.9	ET6 B2 R1-3	22.8 ± 0.9	FT5 B2 R1-3	23.5 ± 0.8
ET1 B3 R1-3	23.6 ± 0.9	ET6 B3 R1-3	22.1 ± 1.0	FT5 B3 R1-3	23.4 ± 0.8
ET2 B1 R1-3	24.8 ± 1.1	FT1 B1 R1-3	4.5 ± 0.2	PT1 B1 R1-3	5.4 ± 0.2
ET2 B2 R1-3	24.6 ± 1.2	FT1 B2 R1-3	4.0 ± 0.2	PT1 B2 R1-3	8.2 ± 0.3
ET2 B3 R1-3	23.4 ± 1.2	FT1 B3 R1-3	4.3 ± 0.2	PT1 B3 R1-3	6.0 ± 0.2
ET3 B1 R1-3	21.0 ± 1.0	FT2 B1 R1-3	14.2 ± 0.6	PT2 B1 R1-3	11.1 ± 0.4
ET3 B2 R1-3	21.5 ± 1.1	FT2 B2 R1-3	22.2 ± 0.7	PT2 B2 R1-3	10.2 ± 0.3
ET3 B3 R1-3	20.7 ± 0.9	FT2 B3 R1-3	20.7 ± 0.6	PT2 B3 R1-3	7.1 ± 0.3
ET4 B1 R1-3	23.6 ± 1.4	FT3 B1 R1-3	9.3 ± 0.3	PT3 B1 R1-3	8.5 ± 0.3
ET4 B2 R1-3	23.9 ± 1.1	FT3 B2 R1-3	8.6 ± 0.3	PT3 B2 R1-3	9.6 ± 0.3
ET4 B3 R1-3	24.7 ± 0.8	FT3 B3 R1-3	9.0 ± 0.3	PT3 B3 R1-3	11.0 ± 0.4
ET5 B1 R1-3	15.2 ± 0.6	FT4 B1 R1-3	19.1 ± 0.7	PT4 B1 R1-3	10.2 ± 0.3
ET5 B2 R1-3	13.8 ± 0.6	FT4 B2 R1-3	22.6 ± 0.7	PT4 B2 R1-3	11.5 ± 0.4
ET5 B3 R1-3	14.2 ± 0.5	FT4 B3 R1-3	21.4 ± 0.7	PT4 B3 R1-3	10.6 ± 0.4

* Sample ID codes: etizolam (E), flubromazepam (F), pyrazolam (P), type of sample shows tablet (T), related batch (B), and number of replicates (R).

**Table 4 pharmacy-12-00013-t004:** Summary of the analytical findings for flubromazepam.

Drug Sample ID *	UHPLC-MS					^1^H-NMR (600 MHz, METHANOL-D4)	
	Drug Compound Retention Time (min)	Molecular Mass (*m*/*z*)	Base Peak (*m*/*z*)	Collection of Molecular ions and adducts	Confirmed ID * Using Literature (Cayman Chemical)	J Couplings Report	Confirmed ID * (Ref Std ** or Literature)
FRS	2.45	333.15	333	333, 335, 336, 355	Flubromazepam	δ 7.69–7.67 (m, 1H), 7.58–7.54 (m, 2H), 7.34–7.27 (m, 2H), 7.18–7.15 (m, 2H), 4.35–4.20 (s, 2H)	Flubromazepam identified
FT1	2.47	333.15	333	333, 335, 336, 357	Flubromazepam	δ 7.67 (dd, J = 8.6, 2.4 Hz, 1H), 7.57–7.53 (m, 2H), 7.32–7.25 (m, 2H), 7.15 (t, J = 9.6 Hz, 2H), 4.26 (s, 2H), 2.17 (t, J = 7.6 Hz, 1H), 1.57 (s, 1H), 0.87 (t, J = 7.2 Hz, 1H)]	Flubromazepam identified
FT2	2.45	333.15	333	333, 335, 336, 357	Flubromazepam	δ 7.66 (dd, J = 8.6, 2.4 Hz, 1H), 7.56–7.53 (m, 2H), 7.32–7.25 (m, 2H), 7.16–7.13 (m, 2H), 4.63 (s, 3H), 4.25 (s, 2H), 3.29 (t, J = 1.7 Hz, 1H), 1.27 (d, J = 17.2 Hz, 3H), 0.87 (d, J = 6.9 Hz, 0H)	Flubromazepam identified
FT3	2.44	333.15	333	333, 335, 336, 357	Flubromazepam	δ 7.67 (dd, J = 8.6, 2.4 Hz, 1H), 7.57–7.53 (m, 2H), 7.32–7.25 (m, 2H), 7.16–7.13 (m, 2H), 4.63 (s, 4H), 4.26 (s, 2H), 2.13 (s, 0H), 1.29–1.26 (m, 5H), 0.87 (t, J = 6.9 Hz, 1H)	Flubromazepam identified
FT4	2.44	333.15	333	333, 335, 336, 357	Flubromazepam	δ 7.66 (dd, J = 8.6, 2.4 Hz, 1H), 7.56–7.53 (m, 2H), 7.32–7.25 (m, 2H), 7.14 (t, J = 9.3 Hz, 2H), 4.63 (s, 2H), 4.25 (s, 2H), 1.27 (d, J = 17.2 Hz, 3H), 0.87 (d, J = 6.9 Hz, 0H)	Flubromazepam identified
FT4	2.45	333.15	333	333, 335, 336, 357	Flubromazepam	δ 7.66 (dd, J = 8.9, 2.1 Hz, 1H), 7.55 (q, J = 6.9 Hz, 2H), 7.32–7.25 (m, 2H), 7.15 (dd, J = 10.3, 8.9 Hz, 2H), 4.63 (s, 3H), 4.25 (s, 2H), 1.26 (s, 2H), 0.87 (s, 0H)	Flubromazepam identified

* ID = identification; ** Ref Std = reference standard.

**Table 5 pharmacy-12-00013-t005:** Summary of the analytical findings for pyrazolam.

Drug Sample ID *	UHPLC-MS					^1^H-NMR (600 MHz, METHANOL-D4)	
	Drug Compound Retention Time (min)	Molecular Mass (*m*/*z*)	Base Peak (*m*/*z*)	Collection of Molecular Ions and Adducts	Confirmed ID * Using Literature (Cayman Chemical)	J Couplings Report	Confirmed ID * (Ref Std ** or Literature)
PRS	1.02	354.2	354	117, 131, 141, 354, 356, 357	Pyrazolam	δ 8.52 (q, J = 2.3 Hz, 1H), 8.08–7.92 (m, 3H), 7.70 (d, J = 8.2 Hz, 1H), 7.54–7.51 (m, 2H), 5.34 (d, J = 13.1 Hz, 1H), 4.34 (d, J = 13.1 Hz, 1H), 2.65 (s, 3H), 1.34 (s, 0H)	Pyrazolam identified
PT1	1.02	354.2	354	117, 131, 141, 354, 356, 357	Pyrazolam	δ 8.50 (d, J = 4.8 Hz, 1H), 8.06–7.91 (m, 3H), 7.69 (d, J = 8.2 Hz, 1H), 7.53–7.49 (m, 2H), 5.32 (d, J = 13.1 Hz, 1H), 4.48 (d, J = 7.6 Hz, 0H), 4.34–4.32 (m, 2H), 2.64 (s, 3H), 2.19–2.14 (m, 3H), 1.90 (s, 0H)	Pyrazolam identified
PT2	1.03	354.2	354	117, 131, 141, 354, 356, 357	Pyrazolam	δ 8.50 (q, J = 2.1 Hz, 1H), 8.06–7.91 (m, 3H), 7.69 (d, J = 8.2 Hz, 1H), 7.53–7.49 (m, 2H), 5.32 (d, J = 12.4 Hz, 1H), 4.32 (d, J = 13.1 Hz, 1H), 2.64 (s, 3H)	Pyrazolam identified
PT3	1.03	354.2	354	117, 131, 141, 354, 356, 357	Pyrazolam	δ 8.50 (d, J = 4.8 Hz, 1H), 8.06–7.92 (m, 3H), 7.69 (d, J = 8.9 Hz, 1H), 7.53–7.49 (m, 2H), 5.32 (d, J = 13.1 Hz, 1H), 4.48 (d, J = 8.2 Hz, 1H), 4.34–4.32 (m, 4H), 2.64 (s, 3H), 1.90 (s, 1H)	Pyrazolam identified
							(weak signal)
PT4	1.02	354.2	354	117, 131, 141, 354, 356, 357	Pyrazolam	δ 8.50 (dd, J = 7.2, 2.4 Hz, 1H), 8.06–7.91 (m, 3H), 7.69 (d, J = 8.9 Hz, 1H), 7.53–7.49 (m, 2H), 5.32 (d, J = 13.1 Hz, 1H), 4.48 (d, J = 7.6 Hz, 1H), 4.34–4.32 (m, 5H), 2.64 (s, 3H), 1.90 (s, 1H)	Pyrazolam identified
							(weak signal)

* ID = identification; ** Ref Std = reference standard.

**Table 6 pharmacy-12-00013-t006:** Themes identified using an automated AI program (ChatGPT); themes are ranked in the order of prevalence. Examples of relevant social media conversations relating to etizolam, flubromazepam, and pyrazolam, as identified by AI, are presented (right). NB: Social media posts may contain spelling mistakes, abbreviations, and informal/street language.

Thematic Analysis Performed by ChatGPT	
Ranked Analysis of Social Media Data	Relevant Social Media Posts Compared to AI Analysis
Drug Use and Experiences: This is the most prevalent theme across the texts. It includes discussions about various drugs, their effects, usage, and poly-substance patterns, and where to purchase them. Users share their experiences, positive outcomes, and concerns about side-effects.	- Pyrazolam doesn’t do anything besides make you not anxious - Etizolam is a benzo, approximately 10 times stronger than prescribed diazepam and frequent use can lead to anxiety, depression and sleep problems - Flubromazepam--help me with dosing. I’m a newb with this chem - Will taking 1000 mg of gabapentin and 3 lyrica and 9 mg of etizolam be dangerous?
2. Harm Reduction and Advice: The second most prevalent theme involves individuals seeking or offering advice on harm reduction, tapering schedules, and minimizing negative effects of drug use. This indicates a concern for safety and responsible drug use.	- Great FLUBROMAZEPAM 4 mg PELLETS. A good replacement for etizolam - I am addicted to bromazolam and planning on tapering off of it with flubromazepam - Would pyrazolam be a good taper benzo from etiz and Xanax?
3. Safety Concerns and Health Effects: Users express concerns about dosing, potential side-effects, withdrawals, and the risks associated with using these substances. There are also discussions about tapering off substances and avoiding cold turkey withdrawals.	- A side-effect of continual pyrazolam use was double vision, it’s such a strong benzo with weak effect - Having a hard time getting off Pyrazolam - Took 2 tablets (0.5 mg) of etizolam tabs that a friend gave to me. Please suggest activities to do and how to ensure harm reduction as it’s my first time consuming this.
4. Comparative Analysis with Other Substances: Many users compare different substances, seeking insights into effects, potency, duration, and potential side-effects. This theme reflects the users’ desire to make informed choices when using substances.	- bromazolam vs. pyrazolam - What are the subjective differences between pyrazolam and bromazolam for you, and which do you prefer? - Is flubromazepam any good? - pyrazolam what’s the duration compared to etizolam?
5. Availability and Sale of Substances: The fifth most prevalent theme revolves around discussions about the availability of various substances, including benzodiazepines and research chemicals. There is an emphasis on sourcing drugs through specific channels and concerns about counterfeit substances and risks associated with purchasing from unverified sources.	- Cheap original #Etizolam #Etizest #Etibest! Register now and use coupon code: 510D to get a −10% discount on your first order - Etizolam CAS 125541-22-2 high quality low price!!! - I’m seeing fake Xannies popping up made with fentanyl so uh, watch out - It’s polydrug hell in Scotland. New category created for “street benzos”, with fake Valium-etizolam-deaths utterly out of control

## Data Availability

Data supporting reported results can be requested from the corresponding author. All X and Reddit posts are available upon request.

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
