# Peer review of "Benzodiazepine Boom: Tracking Etizolam, Pyrazolam, and Flubromazepam from Pre-UK Psychoactive Act 2016 to Present Using Analytical and Social Listening Techniques"

_pharmacy, 2024, doi:10.3390/pharmacy12010013_

Round 1

Reviewer 1 Report

Comments and Suggestions for Authors

Author Response

Reviewer 1

The article aims to build a correspondence between the toxicological analyzes performed on substances purchased on the black market and the data taken from social forums such as X and Reddit. The idea is original. The paper is inspired by toxicological analyzes (UHPLC-MS and NMR) carried out on three different benzodiazepines acquired on the open internet from three separate vendors and relative comparison with forum data.

First criticism: It is not clear how the data was exported to the Google sheet for analysis, how the selection took place thanks to numerous.ai and then how this data was inserted into chatGPT. I ask researchers to show their method accurately step by step. Don't forget that a scientific paper must perfectly describe the method to allow other researchers to replicate it. The results showed the drugs of interest (among Etizolam, Pyrazolam and Flubramezepam, i.e. E-PF) were detected in all the samples with a purity range 53-2450% of the expected ADI. The H NMR could confirm the presence of flubromazepam and pyrazolam but not of etizolam. The netnographic analysis performed from 2016 to 2023 showed that during 2016, Etizolam was the most prevalent drug in term of social media posts and that this trend continued up until 2023. Reddit was the most used social platform if compared to X, while X contained the highest numbers of discussion around etizolam sales and harm.

The authors would like to thank the reviewer for their insightful comments.

  1. Data extraction method and AI analysis:

We appreciate this is a new method and can offer some complexities in understanding the process, therefore we have adjusted the method to reflect detailed analyses in the manuscript.

  1. Those figures are correct and the failure to identify etizolam in the NMR is correct, based on the inability to identify a specific methyl group associated with the azole ring and a three-proton triplet, expected at 1.3 ppm, from the ethyl side chain. This is believed to be due to a level of interference related to this as a street compound and not a pharmaceutical product. Nevertheless, we understand that this is an assumption, and the NMR response is an exact. UHPLC(MS) did however satisfy the identification of etizolam as the main compound.

  1. This is correct, we have discussed that some posts bridged across multiple themes, hence the results.

The second criticism is as follows. The analysis carried out on social media (Reddit and X) developed over a 7-year period from 2016 to 2023, therefore representing a dynamic observatory of how the use of substances E-P-F has changed over time. The analytical study proposed by the Authors, however, was carried out on samples acquired on the black market, probably recently, without being able to compare how the products sold have objectively changed over time. We obtained data on purity and molecular evidence nowadays. Therefore, it may appear that the two lines of the research (social media evaluated over time and analytical analysis evaluated at a specific moment) have little to do one with each other. The authors are therefore asked to explain how this double analysis on different scenarios and different times can find correlation.

  1. We agree that there is overall a lack of quantitative data in terms these types of drugs. The drugs we tested were bought pre the UK PSA 2016, therefore they could be purchased under Home Office License. Post the UK PSA 2016, it is not possible to purchase these drugs under any circumstance, therefore we cannot ‘book-end’ the quantitative analysis. We believe that the ‘nostalgic’ effect of these drugs which became popular before the UK PSA 2019, may still be driving demand today.

Other suggestions in relation to citation that can improve the paper.

In the introduction, in particular on line 95, the Author should add a sentence like this.

“In particolar, adulterants represent a hot topic as on the one hand they can increase both the recreational and harmful effects of the drug, on the other they are subject to changes over time in order to avoid controls by health services or food regulation. [doi.org/10.3390/ijms232314619].

In the discussion, when the authors enter into the merits of the recreational use of etizolam, in particular in the sentence in which they describe the deleterious effects linked to abuse of this substance [on line 651], please cite a relevant paper in which the double face of Etizolam is highlighted such as responsible of a mild sedative effect at low doses (inable to alter driving performance) and a potential lethal agent at high doses. doi.org/10.1016/j.forsciint.2019.05.018

The authors thank the reviewer for their recommendation and have added the relevant papers.

Reviewer 2 Report

Comments and Suggestions for Authors

Benzodiazepine Boom: Tracking Etizolam, Pyrazolam and Flubromazepam from Pre-UK Psychoactive Act 2016 to Present Using Analytical and Social Listening Techniques

The number of benzodiazepines (BZD) used as drugs and sold on the street or online in the European Union is increasing. In this paper, the authors focused on three benzodiazepines (Etizolam, Flubromazepam and Pyrazolam) widespread in the drug market, analyzing (UHPLC-MS(MS) and 1H-NMR) several purchased batches of the three substances over a period of about 7 years. In addition, studies across the platforms X (formerly Twitter) and Reddit, between 2016-2023, on the same drugs were conducted. The high concentration of the three BZD in the batches analysed and frequent posts about the same drugs on the two social (X and Reddit) show that the BZD market presents significant public health concerns due to its over-potent nature and continued popularity.

 Comments

The manuscript is very interesting and presents new and useful contexts. There are no publications in the literature on the subject. The conclusions are consistent with the evidence and the references are appropriate and exhaustive. The tables and the figures provide detailed information. The authors already consider the limitations of this study. English language and style are fine. Only small improvement is recommended as follows:

·      The authors should report that at the 2020 Commission on Narcotic Drugs Sixty-third session, the Commission decided to include etizolam in Schedule IV of the 1971 Convention on Psychotropic Substance. In addition, the corresponding bibliographical reference should be added:

1.     UNODC Laboratory and Scientific Service Portals - March 2020 – UNODC: Twelve substances and one precursor "scheduled" at the 63rd Session of the Commission on Narcotic Drugs

https://www.unodc.org/LSS/Announcement/Details/165b82de-e7ef-4a92-8614-9f8ad4819083

The paper can be accepted after the required revision have been made.

Author Response

Reviewer 2

The manuscript is very interesting and presents new and useful contexts. There are no publications in the literature on the subject. The conclusions are consistent with the evidence and the references are appropriate and exhaustive. The tables and the figures provide detailed information. The authors already consider the limitations of this study. English language and style are fine. Only small improvement is recommended as follows:

  • The authors should report that at the 2020 Commission on Narcotic Drugs Sixty-third session, the Commission decided to include etizolam in Schedule IV of the 1971 Convention on Psychotropic Substance. In addition, the corresponding bibliographical reference should be added:
  1. UNODC Laboratory and Scientific Service Portals - March 2020 – UNODC: Twelve substances and one precursor "scheduled" at the 63rd Session of the Commission on Narcotic Drugs

https://www.unodc.org/LSS/Announcement/Details/165b82de-e7ef-4a92-8614-9f8ad4819083

The authors would like to thank the reviewer for their insightful comment.

  1. The recommended reference has been included with the following sentence; “Etizolam was first reported by the EMCDDA in 2011 and became internationally controlled in 2017 [43-44], prior to its inclusion in Schedule IV of the 1971 Convention on Psychotropic Substances, in 2020 at the Commission on Narcotic Drugs Sixty-third session [45].

Reviewer 3 Report

Comments and Suggestions for Authors

The paper is generally well-written and contains only a few minor grammatical errors such as: lines 401, 425, and 433.  The combination of netnographic analysis and analytical chemistry is an interesting approach. 

There are a large number of tables presently placed one after the other (lines 383 to 388).  This would benefit from being broken up more into the text where they are most relevant.

I was surprised to see that there were no actual NMR spectra in the paper or the relevant appendices, these have only been given as tables.

The axes in figures 2A, 2B, and 2C require labeling and units.

The references in the text are given using surnames and dates, but the citation list uses numbers.  It is very difficult to make judgments on this with the paper in its present form.  This must be corrected in the resubmission.

Comments on the Quality of English Language

few minor errors.

Author Response

Reviewer 3

The paper is generally well-written and contains only a few minor grammatical errors such as: lines 401, 425, and 433.  The combination of netnographic analysis and analytical chemistry is an interesting approach. 

There are a large number of tables presently placed one after the other (lines 383 to 388).  This would benefit from being broken up more into the text where they are most relevant.

I was surprised to see that there were no actual NMR spectra in the paper or the relevant appendices, these have only been given as tables.

The axes in figures 2A, 2B, and 2C require labeling and units.

The references in the text are given using surnames and dates, but the citation list uses numbers.  It is very difficult to make judgments on this with the paper in its present form.  This must be corrected in the resubmission.

The authors would like to thank the reviewer for their insightful comments.

  1. Grammar errors on lines 401, 425, and 433 (full stops and commas) have been amended.
  2. Tables from lines 383 to 388 have now been divided into their corresponding sections, removing the cluster of tables over a number of consecutive pages.
  3. Example NMR spectra has been included to show the response for pyrazolam (P18, line 448-450).
  4. Axes labels and units have been included in figures 2A, B and C as suggested.
  5. All references in text have been provided as numbers, in line with the journal specifications.

Round 2

Reviewer 3 Report

Comments and Suggestions for Authors

This is an interesting report, utilising both a survey of the internet and analytical chemistry.  However, since the authors have now updated the references I found a number of closely related published papers that have not been mentioned.  These need to be included in the submission and commented upon in the text.  How do these papers agree or disagree with the author’s findings?

Anna P. Shapiro, Travis S. Krew, Mohsen Vazirian, Jason Jerry, Christopher Sola, Novel Ways to Acquire Designer Benzodiazepines: A Case Report and Discussion of the Changing Role of the Internet, Psychosomatics, 60, 6, 2019, 625-629, https://doi.org/10.1016/j.psym.2019.02.007.

Abouchedid, R., Gilks, T., Dargan, P.I., Archer, J.R. and Wood, D.M., 2018. Assessment of the availability, cost, and motivations for use over time of the new psychoactive substances—benzodiazepines diclazepam, flubromazepam, and pyrazolam—in the UK. Journal of Medical Toxicology14, pp.134-143.

Lyphout, C., Yates, C., Margolin, Z.R., Dargan, P.I., Dines, A.M., Heyerdahl, F., Hovda, K.E., Giraudon, I., Bucher-Bartelson, B. and Green, J.L., 2019. Presentations to the emergency department with non-medical use of benzodiazepines and Z-drugs: profiling and relation to sales data. European journal of clinical pharmacology, 75, pp.77-85.

Ramos, T.B., Bokehi, L.C., Oliveira, E.B.D., Gomes, M.D.S.A., Bokehi, J.R. and Castilho, S.R.D., 2020. Information about benzodiazepines: what does the internet offer us?. Ciência & Saúde Coletiva, 25, pp.4351-4360.

COLEMAN, J.J., 2020. Benzodiazepines Today and Tomorrow. The Benzodiazepines Crisis.

Littlejohn, C., Baldacchino, A., Schifano, F. and Deluca, P., 2005. Internet pharmacies and online prescription drug sales: a cross-sectional study. Drugs: education, prevention and policy, 12(1), pp.75-80.

Author Response

The authors would like to thank the reviewer for their advice and insight. The recommended references were included and added to the discussion of the manuscript.
